# Adsorptive Behavior of Cu^2+^ and Benzene in Single and Binary Solutions onto Alginate Composite Hydrogel Beads Containing Pitch Pine-Based Biochar

**DOI:** 10.3390/polym14173468

**Published:** 2022-08-25

**Authors:** Saerom Park, Jeong Woo Lee, Ji Eun Kim, Gwangnam Kang, Hyung Joo Kim, Yong-Keun Choi, Sang Hyun Lee

**Affiliations:** 1Department of Biological Engineering, Konkuk University, Seoul 05029, Korea; 2R&D Center, ATE Corporation, Seoul 05029, Korea

**Keywords:** alginate, biochar, composite hydrogel, adsorption, Cu^2+^, benzene

## Abstract

In this study, we prepared alginate composite hydrogel beads containing various compositions of biochar produced from pitch pine (*Pinus rigida*) for the removal of Cu^2+^ and benzene from model pollutant solutions. The properties of the alginate/biochar hydrogel beads were evaluated using scanning electron microscopy, Fourier transform infrared spectroscopy, and Brunauer–Emmet–Teller analyses. Adsorption behavior of alginate/biochar hydrogel beads indicated that the adsorption capacities for Cu^2+^ (28.6–72.7 mg/g) were enhanced with increasing alginate content, whereas the adsorption capacities for benzene (20.0–52.8 mg/g) were improved with increasing biochar content. The alginate/biochar hydrogel beads exhibited similar adsorption capacities for Cu^2+^ and benzene in the concurrent system with Cu^2+^ and benzene compared to those in a single pollutant system. Adsorption kinetics and isotherm studies of the alginate/biochar hydrogel beads followed the pseudo-second-order model (r^2^ = 0.999 for Cu^2+^, and r^2^ = 0.999 for benzene), and Langmuir model (r^2^ = 0.999 for Cu^2+^, and r^2^ = 0.995 for benzene). In addition, alginate/biochar hydrogel beads (containing 1 and 4% biochar) exhibited high reusability (>80%). Therefore, alginate/biochar hydrogel beads can be applied as adsorbents for the removal of multiple pollutants with different properties from wastewater.

## 1. Introduction

Rapid industrialization in primary industries and increasing industrial activities owing to exponential population growth have exacerbated environmental contamination by the emission of various organic and inorganic pollutants such as heavy metals, dyes, pharmaceuticals, antibiotics, herbicides, and volatile organic compounds [1]. These soil and water contaminants have negative long-term effects on human health and the ecosystem. Therefore, it is important to remove harmful contaminants via remediation techniques such as separation, degradation, precipitation, adsorption, and electrolysis [2]. Among these methods, adsorption is one of the most commonly used decontamination methods for aqueous solutions owing to its low cost, convenience, and adaptability [3,4]. However, typical commercial adsorbents such as synthetic resins and activated carbon, are expensive, non-biodegradable, and energy-intensive. Accordingly, cost-effective, eco-friendly, and efficient substitutes must be developed for the removal of pollutants. In particular, biomass derivatives are one of the candidates considered as alternative materials for conventional adsorbents owing to their recyclable, biodegradable, and available features. [5,6,7].

Recently, biochar has attracted attention as an alternative adsorbent owing to its low cost, eco-friendly properties, and physicochemical characteristics with various functional groups. Biochar is a carbonic material produced through the thermochemical conversion of sustainable biomass such as weeds, leaves, vermicompost, algae, sludge, sawdust, and manure [2,8,9,10,11,12,13,14,15], under limited oxygen conditions [16]. Biochar can remove organic and inorganic micropollutants from wastewater owing to its functional groups and carbon-enriched structure [17]. The physicochemical properties of biochar, such as specific surface area, porous structure, functional groups, mineral content, and aromatic structure, are affected by the type of feedstock and production conditions [1,16,18]. Song et al. (2021) reported the effect of feedstock type and pyrolysis temperature on the adsorption capacity of biochar for microcystin-LR. Biochar produced under the same conditions exhibited different adsorption abilities depending on the original biomass, and the adsorption capacity was enhanced by increasing the pyrolytic temperature [19]. In the contaminant reduction process, biochar can be utilized as an adsorbent for the immobilization of various organic and inorganic pollutants, such as dyes, antibiotics, volatile organic compounds, pharmaceuticals, and heavy metals [9,10,11,17,20,21]. However, the small particle size of biochar causes the migration of adsorbed micropollutants into the water and soil, resulting in accidental contamination [22].

Alginate, a biopolymer extracted from brown algae, is a linear polysaccharide composed of two monomeric units: a-L-guluronate (G) and b-D-mannuronate (M). Based on the ratio of the G and M blocks, alginate forms a water-insoluble and thermo-irreversible gel via ionic crosslinking with polyvalent cations or acid treatment [23,24]. In particular, alginate has abundant hydroxyl and carboxyl groups and can be used as a potential adsorbent for decontamination [3,25]. In several studies, calcium alginate beads have been employed to remove various micropollutants, such as heavy metals, dyes, and inorganic ions [26,27,28,29,30]. However, several drawbacks of calcium alginate beads, such as mechanical strength, stability, thermal resistance, low porosity, and high density, must be overcome for industrial applications [3,23,25]. Therefore, modification techniques have been applied to improve alginate performance. Chemical modification is conducted through chemical reactions, such as oxidation, amination, sulfation, and esterification of hydroxyl or carboxyl groups of alginate, to introduce novel functional groups [31]. Blending alginate and additives, such as polymers, organic compounds, and inorganic materials, are physical functionalization methods for the fabrication of alginate-based adsorbents [23].

Alginate composites incorporating carbonic materials, such as activated carbon, carbon nanotubes, graphene oxide, and biochar, exhibited enhanced specific surface area, porosity, and mechanical strength compared to calcium alginate beads. Moreover, alginate/biochar composites provide a network structure for leakage prevention of fine-sized carbon-rich solids and can be easily separated from liquid treatment systems [3,23,25]. Therefore, alginate/biochar composite materials can be used for the remediation of contaminants, such as heavy metals, dyes, volatile organic compounds, antibiotics, pharmaceuticals, and inorganic ions [32,33,34,35,36].

In this study, alginate/biochar composite hydrogel beads were prepared using pitch pine-based biochar obtained through pyrolysis. The physical and chemical properties of the alginate/biochar hydrogel beads were characterized. The correlation between the compositions (alginate and biochar) of the beads and their adsorption capacities for Cu^2+^ and benzene was investigated. The adsorptive behaviors of Cu^2+^ and benzene onto the alginate/biochar hydrogel beads were also analyzed using various kinetic and isotherm models. Furthermore, the Cu^2+^ and benzene adsorption capacities of the alginate/biochar hydrogel beads in a single contaminant solution were compared with those of the binary solution. Moreover, the reusability of the alginate/biochar hydrogel beads as adsorbents was evaluated in this study.

## 2. Materials and Methods

### 2.1. Materials

Sodium alginate, calcium chloride, benzene, copper sulfate, nitric acid, ethanol, and HPLC-grade water were purchased from Samchun Pure Chemical Co., Ltd. (Gyeonggi-do, Republic of Korea). Sodium citrate was purchased from Deajung Chemical & Metals Co., Ltd. (Gyeonggi-do, Republic of Korea). Wood powder from pitch pine (*Pinus rigida*) was obtained from G-Biotech (Seoul, Republic of Korea). All the other chemicals used in this study were of analytical grade and used without further purification.

### 2.2. Preparation of Pitch Pine-Based Biochar and Alginate Hydrogel Beads

To produce pitch pine-based biochar, wood powder was pyrolyzed for 2 h under a nitrogen environment at different temperatures (350, 550, and 750 °C) using a pyrolysis reactor (Tube Furnace OTF-1200X-S, MTI Co., Richmond, CA, USA). The obtained biochar was washed several times with distilled water (DW) to remove ash and impurities and then dried at 60 °C overnight.

For the preparation of adsorbents in the hydrogel state, 4% (*w*/*w*) sodium alginate and various contents of biochar (0–10%, *w*/*w*) were mixed in DW using a mortar and pestle. The alginate solution containing homogeneously dispersed biochar was added dropwise to a 2% CaCl_2_ solution at a rate of 50 mL/min with a syringe pump (LSP01-2A; Longer Pump, Baoding City, China) using an 18-gauge needle to obtain millimeter-sized hydrogel beads. The resulting hydrogel beads were cured in a CaCl_2_ solution for 1 h and then washed with DW three times. The prepared alginate (Alg) and alginate/biochar (Alg/BC) hydrogel beads were stored in DW at 4 °C until further use.

### 2.3. Characterization of Pitch Pine Powder, Biochar, and Alg/BC Beads

Thermogravimetric analysis (TGA) was conducted from 25 to 900 °C at a rate of 20 °C/min under N_2_ using a TGA N-1000 instrument (Scinco Co., Ltd., Seoul, Republic of Korea). The Brunauer–Emmett–Teller (BET) specific surface area was investigated by N_2_ sorption using TriStar II (Micromeritics, Norcross, GA, USA). Fourier transform infrared spectroscopy (FTIR) measurements were performed by averaging 16 scans per spectrum in the range of 4000–700 cm^−1^ (resolution, 4 cm^−1^; scanning interval, 1 cm^−1^) using an FT/IR-4100 type A spectrometer (Jasco International Co., Ltd., Tokyo, Japan). The surfaces of the samples were examined using scanning electron microscopy (SEM, TM4000 Plus, Hitachi Co., Tokyo, Japan). The functional groups and morphology of the Alg/BC hydrogel beads were analyzed by FTIR and SEM after being frozen at −70 °C and dried at −80 °C under vacuum. The biochar content of the Alg/BC hydrogel beads was measured by dissolving alginate in 100 mM sodium citrate at 60 °C for 24 h. The remaining biochar was weighed after washing with DW and drying at 60 °C.

### 2.4. Adsorption of Cu^2+^ and Benzene onto Adsorbents

10 mg (dry weight) of the adsorbent, that is, biochar or Alg/BC hydrogel beads, was added to 10 mL of CuSO_4_ solution (100 mg/L as Cu^2+^). For benzene adsorption experiments, 10 mg (dry weight) of adsorbent was placed in a 5 mL glass vial and then 4.5 mL of benzene solution (300 mg/L) was added. Batch adsorption experiments were conducted by shaking at 120 rpm for 24 h at 25 °C. After each adsorption reaction, for Cu^2+^ measurement, aliquots (200 μL) were periodically collected and centrifuged before dilution with 800 μL of DW. To determine the benzene concentration, aliquots (1 mL) were filtrated using PVDF syringe filter (0.22 μm). Then, the remaining Cu^2+^ and benzene concentration in samples were analyzed. The amounts of adsorbed Cu^2+^ and benzene were measured using a UV/Vis spectrophotometer. The Cu^2+^ content in the solution was measured using Micro BCA^TM^ Protein Assay Kits (Thermo Fisher Scientific, Waltham, MA, USA) according to a modified protocol. BSA (Bovine Serum Albumin) solution (10 mg/mL) was added into BCA reagents instead of CuSO_4_ solution. The prepared BCA working reagents were mixed with the sample and incubated at 60 °C for 1 h. The reacted samples were measured at 562 nm. The benzene concentration was quantified using standard curve (y = 1.508 × 10^−3^x − 6.983 × 10^−3^, R^2^ = 0.999) at 254 nm. The adsorption efficiency (q_e_) of the adsorbent was calculated as:q_e_ = (C_0_ − C_e_)/m × V,(1)
where C_0_ and C_e_ (mg/L) denote the initial and equilibrium concentrations of the adsorption solution, respectively; V (L) denotes the volume of the adsorption solution; and m (g) denotes the weight of the dried adsorbent.

To evaluate the effect of solution pH on Cu^2+^ and benzene adsorption onto the Alg/BC hydrogel beads, the pH of the solution was adjusted between 3 and 6 with 0.1 M HCl or 0.1 M NaOH. The co-adsorption of Cu^2+^ (100 mg/L) and benzene (300 mg/L) onto the Alg/BC hydrogel beads was performed by shaking at 120 rpm for 24 h at 25 °C and pH 6.

### 2.5. Modeling of Adsorption Kinetics and Isotherms

Adsorption kinetics studies for Cu^2+^ and benzene on Alg/BC hydrogel beads were implemented in Cu^2+^ (100 mg/L) and benzene (300 mg/L) solutions with 10 mg (dry weight) of Alg/BC hydrogel beads at 120 rpm and 25 °C. The adsorption isotherm studies were conducted in Cu^2+^ (10–800 mg/L) and benzene (50–800 mg/L) solution for 24 h. Adsorption kinetics and isotherm models were applied to elucidate the results from the adsorption and isotherm experiments using the equations shown in Appendix A.

### 2.6. Reusability of Alg/BC Hydrogel Beads

The adsorbed Cu^2+^ was desorbed from the adsorbents using 0.1 M HCl as the desorption solvent at 120 rpm for 2 h at 25 °C. After the desorption process, the Alg/BC hydrogel beads were regenerated with 2% CaCl_2_ for 2 h and then washed with DW. Benzene desorption from the adsorbents was conducted using ethanol at 120 rpm for 24 h at 25 °C. The adsorbents with desorbed benzene were washed three times with DW.

### 2.7. Statistical Analysis

The triplicate data were analyzed by *t*-test or one-way ANOVA with Tukey’s HSD test using Minitab 16.0 software (Minitab Inc., State College, PA, USA). Statistical significance was set at a 95% confidence interval.

## 3. Results and Discussion

### 3.1. Characterization of Pitch Pine-Based Biochar

Appendix A shows the TGA thermogram of the pitch pine powder from 25 to 900 °C. First, moisture was vaporized at 100 °C, and then the initial mass change was observed between 200 and 380 °C. Because hemicellulose and cellulose start to decompose above 200 and 300 °C, respectively, the sharp decomposition indicates the decomposition of hemicellulose and cellulose [37,38]. Therefore, it is assumed that the main composition of the biochar obtained at 550 and 750 °C is derived from lignin, and the biochar pyrolyzed at 350 °C is thought to originate from lignin and some cellulose. Based on the TGA results, biochars were prepared at different pyrolysis temperatures, that is, 350, 550, and 750 °C, using pitch pine powder. As shown in Appendix A, the biochar yield decreased from 35.6 to 17.0% with increasing pyrolysis temperatures. These results imply that a higher pyrolytic temperature causes the decomposition of cellulose, hemicellulose, and lignin, and the generation of tar and syngas, resulting in a decline in the biochar production yield [39].

An increase in the pyrolysis temperature enhanced the formation of pores on the surface of the biochar (Appendix A). The surface area of the biochar produced at 550 °C was larger than that produced at 350 °C, whereas the biochar produced at 750 °C had a smaller surface area than that produced at 550 °C (Appendix A). These phenomena may indicate that high temperatures can break down the porous structures of biochar and plug the pores with tar produced through the biomass pyrolysis process, leading to a decrease in the surface area of the biochar [19,40].

Appendix A shows the FTIR spectra of biochar prepared at different pyrolysis temperatures. The spectrum of biochar produced at 350 °C shows several peaks at 875 (CO_3_^2−^), 1049 (C-O stretching), 1270 (C-OH phenolics), 1430 (C-H bending), 1640 (aromatic conjugated C=O, C=C bond), 1724 (C=O stretching of carboxyl), 2930 (CH stretching of CH_2_, CH_3_), and 3200–3400 cm^−1^ (-OH band) [2,11,19,20]. However, when the pyrolysis temperature increased, the peaks of the biochar spectra dissipated. According to a previous study, these results suggest that functional groups typically disappear at pyrolysis temperatures above 500 °C [2,19,20].

To evaluate the adsorption capacities of the pitch pine-based biochars produced at 350, 550, and 750 °C for heavy metals, BTEX, Cu^2+^, and benzene were selected as adsorbate models. The adsorption capacity of Cu^2+^ on biochar decreased with increasing pyrolysis temperature (*p* < 0.05), whereas the amount of adsorbed benzene on biochar exhibited the highest value (60.4 mg/g) when using biochar obtained through pyrolysis at 550 °C (*p* < 0.05) (Figure 1). These results indicate that the adsorption capacity of biochar for Cu^2+^ and benzene was influenced by the contents of various functional groups (hydroxyl and carboxyl) and the higher specific surface area (SSA) of biochar (that is, SSA of biochar produced at 550 °C was 297.4 m^2^/g) (Appendix A and Appendix A). All biochars showed trivial adsorption capacities for Cu^2+^ (<3 mg/g) compared to that of benzene (>25 mg/g). Thus, the biochar generated at 550 °C was only appropriate for the adsorption of benzene. Hence, the biochar prepared in this study was unsuitable for adsorbing heavy metals such as Cu^2+^ without chemical activation or modification processes. Therefore, in the following experiments, alginate (a negatively charged biopolymer) was blended with pitch pine-based biochar obtained at 550 °C to improve its heavy metal adsorption capacity.

### 3.2. Characterization of Alg/BC Beads

The 4% alginate solutions containing various amounts of biochar were made into hydrogel beads of regular spherical shape by the extrusion method using an 18-gauge needle; however, the Alg/BC solution containing more than 10% biochar formed irregular beads. The alginate beads were pale yellow, whereas the addition of biochar to the alginate solution changed them to black (Figure 2). Furthermore, the surface of the Alg/BC hydrogel beads exhibited significant changes depending on the biochar content. The alginate beads displayed a smooth surface, whereas the surface of the Alg/BC beads was rough with micro-sized bumps owing to the irregular pore structure of the biochar. The average size and weight of the dried alginate beads were 1.2 mm and 0.7 mg, respectively. The addition of biochar to the alginate solution caused a significant increase in the average particle size and weight (Appendix A). The average size and weight of the dried Alg/BC beads prepared with 4% alginate and 10% biochar were 2.7 mm and 3.8 mg, respectively.

### 3.3. Adsorption Capacities of Alg/BC Hydrogel Beads for Cu^2+^ and Benzene

Figure 3 shows the adsorption capacities of the Alg/BC hydrogel beads for Cu^2+^ and benzene based on the added biochar content. The adsorption capacity of the alginate hydrogel beads for Cu^2+^ was 54 times higher than that of the biochar generated at 550 °C. The increase in the added biochar content caused a decrease in the Cu^2+^ adsorption capacity, whereas the adsorption capacity of benzene increased with an increase in biochar. Furthermore, good linear correlations were observed between added biochar content and adsorption capacity (Cu^2+^: r^2^ = 0.984, and benzene: r^2^ = 0.968). Thus, these results indicated that the adsorption capacities of the Alg/BC hydrogel beads for Cu^2+^ and benzene were highly dependent on the ratios of alginate and biochar content in the Alg/BC hydrogel beads. Sigdel et al. (2017) showed that powdered activated carbon-impregnated alginate beads exhibited a similar phenomenon for the adsorption of heavy metal ions and aromatic compounds owing to the physicochemical characteristics of adsorbents. Hence, the ratios of alginate and biochar contents can be selectively applied depending on the concentration and types of contaminants in the binary solution.

The FTIR spectra of the alginate and Alg/BC beads (prepared with 4% alginate and 4% BC) before and after adsorption are shown in Appendix A. There are various peaks at 1030 (C-C stretching), 1420 (symmetric vibration of COO^−^), 1600 (asymmetric stretching vibration of COO^−^), and 3200–3400 cm^−1^ (-OH band), corresponding to the functional groups of alginate [36,41,42,43]. In addition, there were no significant changes of various peaks before and after Cu^2+^ and benzene adsorption. Nevertheless, some changes (shift and/or intensity) may occur after Cu^2+^ adsorption as reported form the previous literature [42]. Mohammed et. al. (2018) reported that the interaction between the divalent ion (e.g., Pb^2+^, Cu^2+^, and Cd^2+^) and carboxylate group leads the ability to bind [42].

### 3.4. Effect of Initial Solution pH on the Adsorption of Cu^2+^ and Benzene

The initial pH of the solution influences the adsorptive interactions between the adsorbates and adsorbents. Hence, adsorption experiments for Cu^2+^ and benzene on alginate and Alg/BC hydrogel beads (prepared with 1 and 4% biochar) were conducted in the pH range of 3 to 6. The adsorption capacity for Cu^2+^ of hydrogel beads consisting of alginate and biochar increased with increasing solution pH (one-way ANOVA, *p* < 0.05) (Figure 4a). These phenomena may be caused by competition between divalent metal ions and the high concentration of protons at low pH for adsorption sites [43,44]. When the pH value of the liquid phase is above 4.7, the metal ions (that is, positive charge) adsorb onto the negatively charged carboxyl groups of alginate via electrostatic interactions [4,35]. Therefore, these results suggest that the adsorption of Cu^2+^ onto alginate and Alg/BC hydrogel beads progressed mainly through electrostatic interactions. In contrast, as shown in Figure 4b, the initial solution pH had no significant effect on the adsorption of benzene onto alginate and Alg/BC hydrogel beads (one-way ANOVA, *p* > 0.05). Sigdel et al. (2017) reported that the solution pH had little effect on the sorption of benzene by alginate beads containing activated carbon. These results imply that benzene adsorption onto the alginate and Alg/BC hydrogel beads progressed through non-electrostatic interactions (that is, hydrogen bonding, π–π interactions, and hydrophobic interactions) in addition to association with functional groups [20].

### 3.5. Adsorption Kinetics Study for Cu^2+^ and Benzene onto Alg/BC Hydrogel Beads

Figure 5a,c show the adsorption of Cu^2+^ and benzene onto the adsorbents based on the contact time, respectively. The adsorption of Cu^2+^ onto all adsorbents rapidly occurred within 8 h and then plateaued (one-way ANOVA, *p* > 0.05). In particular, the alginate hydrogel beads exhibited 1.7 times higher adsorption capacity for Cu^2+^ than that of Alg/BC hydrogel beads (4% biochar) owing to the higher content of negatively charged carboxyl groups of alginate compared to Alg/BC hydrogel beads (Figure 5a). In contrast, the benzene adsorption process required more than 16 h to reach the equilibrium state (one-way ANOVA, *p* > 0.05), and the amount of benzene adsorbed onto the Alg/BC hydrogel beads (4% biochar) increased by up to 3.3 times compared to the pure alginate beads (Figure 5c). These results suggest that the ratios of alginate and biochar significantly influence the adsorption of Cu^2+^ and benzene, as indicated by the above results, indicating chemical and physical adsorption. In addition, alginate provides oxygen in various functional groups (such as carboxyl and hydroxyl groups) for cross-linking via ion exchange between divalent metal ions and adsorbents to form an egg-box structure. Additionally, negatively charged alginate draws positively charged metal ions via electrostatic interactions [3,35,42]. In contrast, functional groups with abundant oxygen in alginate disrupt the adsorption of hydrophobic benzene onto Alg/BC hydrogel beads [21].

Typically, the adsorption process consists of multiple mass transfer steps: (1) migration of the adsorbate from the bulk solution to the boundary layer of the adsorbent; (2) external mass transfer between the boundary layer and the surface of the adsorbent (boundary layer and liquid film diffusions); (3) intraparticle diffusion within the adsorbent (pore and surface diffusions); and (4) adsorption of the adsorbate on the active site of the adsorbent via physisorption or chemisorption [45,46,47,48]. To elucidate the mechanism of Cu^2+^ and benzene adsorption onto the Alg/BC hydrogel beads, adsorption kinetics studies were performed using pseudo-first-order, pseudo-second-order, Elovich, intraparticle diffusion, and liquid film diffusion models. The pseudo-first-order model was used to describe the pure physical adsorption. The pseudo-second-order model indicates that the rate-limiting step is the chemical or physicochemical adsorption. The Elovich model has been applied to the chemisorption of adsorbates on adsorbents [43,45,49,50]. As shown in Figure 5b,d and Table 1, the adsorption of Cu^2+^ and benzene onto alginate and Alg/BC hydrogel beads followed a pseudo-second-order kinetic model with high correlation values (r^2^ > 0.99), and the calculated q_e_ values were close to those obtained experimentally. These results show that the adsorption of Cu^2+^ and benzene onto alginate and Alg/BC hydrogel beads was governed by electrostatic interactions and chemisorption [51]. In a previous study, the process of Zn^2+^ adsorption using alginate beads containing biochar obtained from pine cones was fitted to the pseudo-second-order model with a high regression coefficient (r^2^ > 0.99), similar to the results of this study [52]. Gürkan et al. (2021) and Roh et al. (2015) reported similar results for alginate composites involving biochar prepared from rice husk and buffalo weed in the divalent metal ion adsorption process [12,43]. In particular, Sigdel et al. (2017) reported that the pseudo-second-order model correlated with the adsorption of Cd^2+^ and benzene onto alginate-activated carbon beads [34].

The intraparticle and liquid film diffusion models suggest the rate-determining step by investigating whether their linear plots pass through the origin. Appendix A shows the plots of the diffusion kinetics models for the adsorption of Cu^2+^ and benzene onto the alginate and Alg/BC hydrogel beads. It is evident that the qt vs. t^0.5^ plots were composed of two linear portions, indicating the boundary and intraparticle diffusions, respectively. However, the plots obtained using diffusion models did not pass through the origin. Hence, in the adsorption of alginate and Alg/BC hydrogel beads for Cu^2+^ and benzene, neither intraparticle diffusion nor liquid film diffusion was the sole rate-controlling step, but these may be simultaneous adsorption rate-controlling steps [47,53].

### 3.6. Adsorption Isotherm Study for Cu^2+^ and Benzene onto Alg/BC Hydrogel Beads

The adsorption capacities of alginate and Alg/BC hydrogel beads increased with increasing initial concentration of Cu^2+^ and benzene. Although the adsorption capacities of alginate hydrogel beads for Cu^2+^ and benzene were 130.9 and 21.5 mg/g, respectively, the adsorption capacities of Alg/BC hydrogel beads (prepared with 4% alginate and 4% biochar) for Cu^2+^ and benzene were 57.5 and 74.9 mg/g, respectively (Figure 6a,c). As shown in Figure 3, the proportion of alginate and biochar in the composite hydrogel beads affects the adsorption capacity for Cu^2+^ and benzene. Thus, these results could be due to a change in the bead composition, resulting in alterations in the quantity of active sites for adsorption.

The parameters obtained from the isotherm models are listed in Table 2. The Cu^2+^ and benzene adsorption processes were found to be Langmuir-possessing, which showed the highest correlation (r^2^ > 0.99) (Figure 6b,d). Hence, alginate and Alg/BC hydrogel beads adsorbed Cu^2+^ and benzene on their homogenous surfaces via monolayer adsorption. In the Langmuir equation, the Langmuir constant (K_L_) indicates surface area and porosity correlated with adsorption capacity of adsorbents. Thus, the dimensionless separation factor (R_L_) acquired from the Langmuir model means the favorability of adsorption. The isotherm type is divided into unfavorable (R_L_ > 1), linear (R_L_ = 1), favorable (0 < R_L_ < 1), and irreversible (R_L_ = 0), depending on the R_L_ value. In this study, the R_L_ values for Cu^2+^ and benzene adsorption onto alginate and Alg/BC hydrogel beads ranged between 0 and 1, and thus the adsorption of Cu^2+^ and benzene on alginate and Alg/BC hydrogel beads is a favorable process.

The maximum adsorption capacity, q_m_, of alginate hydrogel beads for Cu^2+^ and benzene was 122.0 and 22.2 mg/g, respectively. The Alg/BC hydrogel beads (prepared with 4% alginate and 4% BC) exhibited 3.7 times higher maximum adsorption capacity for benzene than that of alginate hydrogel beads. In contrast, the q_m_ of the Alg/BC hydrogel beads (prepared with 4% alginate and 4% BC) for Cu^2+^ decreased to 62.5 mg/g. Hence, the addition of biochar to Alg/BC hydrogel beads decreased the q_m_ of the adsorbent for Cu^2+^ while increasing that for benzene adsorption.

### 3.7. Simultaneous Adsorption of Cu^2+^ and Benzene onto Alg/BC Hydrogel Beads

To confirm the interference effect between Cu^2+^ and benzene during multicomponent adsorption onto Alg/BC hydrogel beads (prepared with 4% alginate and 4% biochar), the adsorption reaction was conducted using a binary solution involving both contaminants (Cu^2+^ and benzene). Figure 7 shows the adsorption capacities of the Alg/BC hydrogel beads for Cu^2+^ and benzene in single and binary solutions. The adsorption capacity of Alg/BC hydrogel beads in single-sorbate solution for Cu^2+^ and benzene was 68.6 and 25.9 mg/g, respectively. In a simultaneous adsorption system, the adsorption capacity of Alg/BC hydrogel beads for Cu^2+^ and benzene was 70.8 and 21.8 mg/g, respectively. These results indicated that there was no significant difference in the adsorption ability (*p* > 0.05). Sigdel et al. (2017) reported similar results for cadmium and benzene adsorption on alginate/activated carbon beads [34]. In the binary contaminant solution, there was no competition between Cd and benzene. This indicates that divalent metal ions with positive charges are adsorbed on the active sites of alginate, whereas benzene is adsorbed on the carbon moiety of the adsorbents. Hence, adsorbents composed of alginate and carbonic materials can simultaneously adsorb multi-components with no deterrent effect owing to their different adsorption sites.

### 3.8. Reusability of Alg/BC Hydrogel Beads

To analyze the reusability of Alg/BC hydrogel beads, they were prepared with 1 and 4% BC and used for the removal of Cu^2+^ and benzene, respectively. Figure 8 shows the adsorption and desorption abilities of Alg/BC hydrogel beads. During the adsorption process of Cu^2+^, the residual adsorption capacity of the Alg/BC hydrogel beads was 99.1% of the initial adsorption capacity after five cycles. When a 0.1 M HCl solution was used as the desorption solvent, the desorption efficiency of Cu^2+^ adsorbed on the adsorbents was more than 95.5% of the adsorbed amount. These results may be due to the regeneration of active sites through ion exchange during the desorption and reconstruction processes [3]. The desorption efficiency of benzene from the Alg/BC hydrogel beads was lower than that of Cu^2+^ and ranged from 79.1 to 88.4% for the adsorbed benzene. However, the residual adsorption capacity of the Alg/BC hydrogel beads after reuse was similar to or higher compared with the adsorption efficiency of Cu^2+^ after repeated operation. These results suggest that the Alg/BC hydrogel beads can be used repeatedly during the desorption process using an appropriate desorption solvent for Cu^2+^ or benzene.

## 4. Conclusions

In this study, to obtain sustainable adsorbents, pitch pine-based biochar was blended with alginate and fabricated into hydrogel beads. The Alg/BC composite hydrogel beads showed different adsorption capacities for Cu^2+^ and benzene, based on the composition of alginate and biochar. Biochar and alginate had positive effects on the adsorption capacities of benzene and Cu^2+^, respectively. Furthermore, the initial solution pH influenced the adsorption of Cu^2+^, whereas its effect on the adsorption capacity for benzene was negligible. The pseudo-second-order and Langmuir models were the best-fitting kinetic and isotherm models for the adsorption process on the Alg/BC hydrogel beads, respectively. The Alg/BC hydrogel beads exhibited a trivial difference in adsorption capacities between the single and binary contaminant systems. Additionally, the decrease in the adsorption efficiency of Cu^2+^ and benzene onto the Alg/BC hydrogel beads was trivial even after four reuses. Hence, the Alg/BC hydrogel beads could be customized to be optimized for the treatment of wastewater containing a variety of contaminants, such as heavy metals, dyes, pharmaceuticals, antibiotics, and toxic organic compounds.

## Figures and Tables

**Figure 1 polymers-14-03468-f001:**
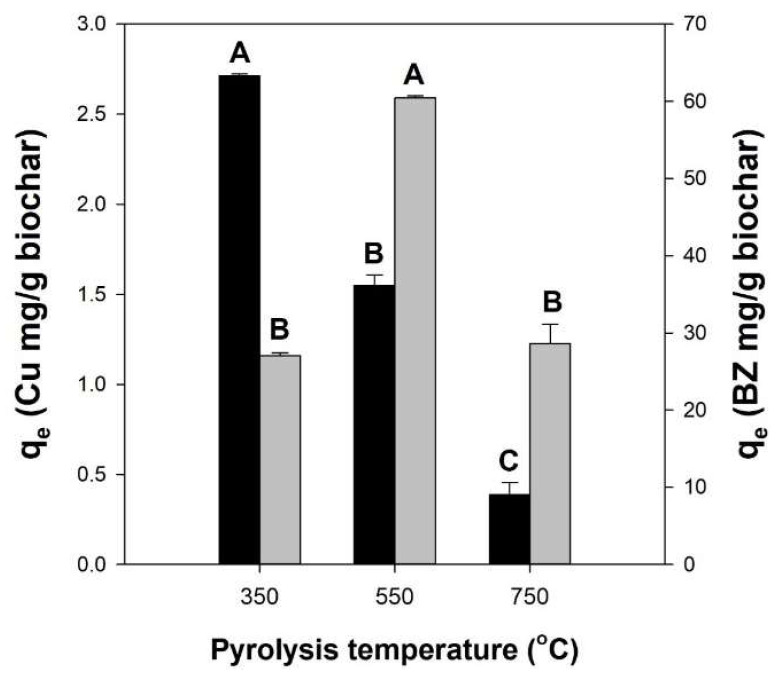
Adsorption capacities of pitch pine-based biochars produced at different pyrolytic temperatures (350, 550, and 750 °C) for Cu^2+^ (black bars) and benzene (gray bars) at equilibrium. One-way ANOVA with Tukey’s test (*p* < 0.05); A, B, and C indicate group classified from Tukey’s test. The same letter means that there is no significant difference between the data.

**Figure 2 polymers-14-03468-f002:**
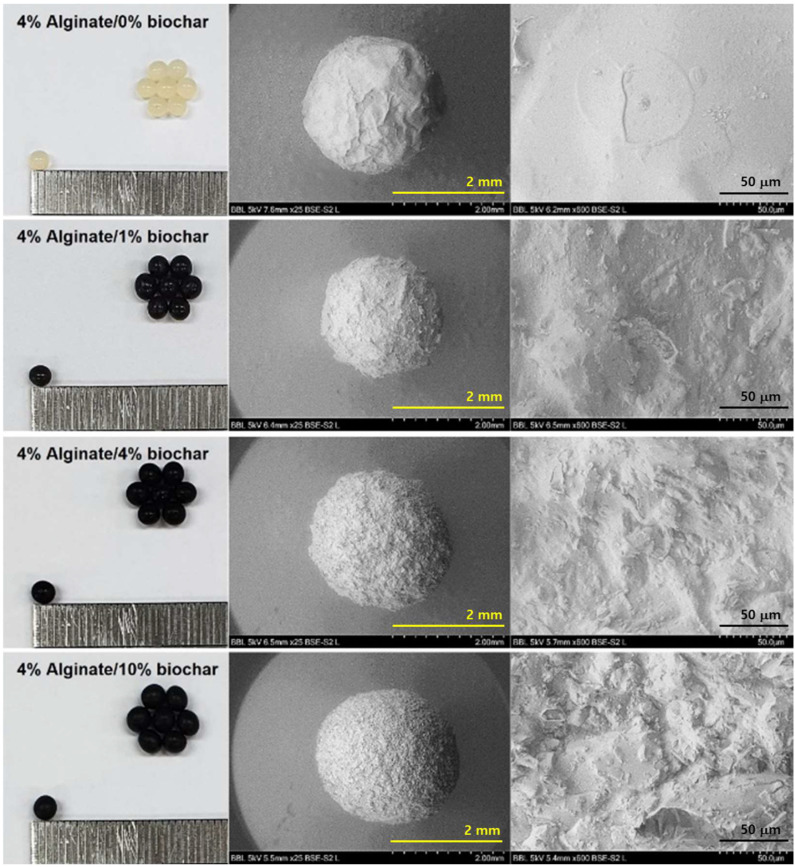
Optical microscopy and SEM images of alginate and alginate/biochar hydrogel beads.

**Figure 3 polymers-14-03468-f003:**
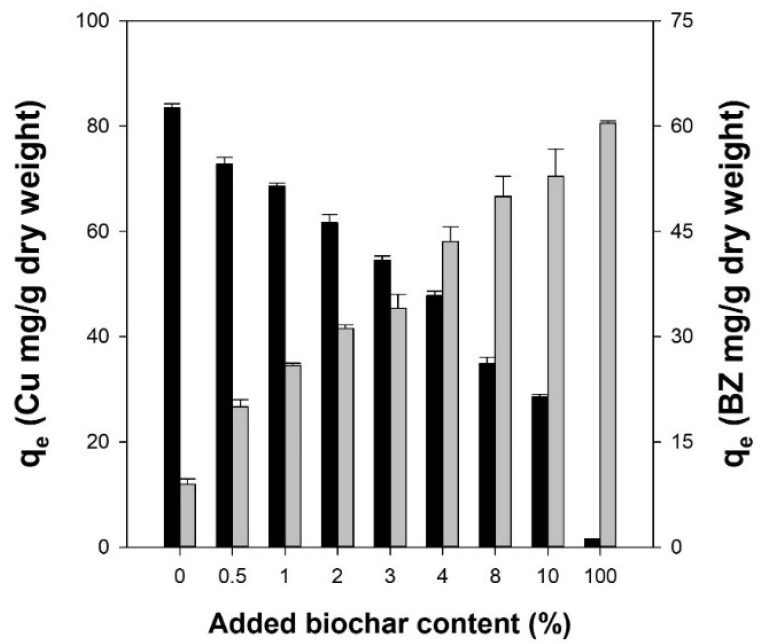
Adsorption capacities of alginate hydrogel beads, alginate/biochar hydrogel beads, and biochar for Cu^2+^ (black bars) and benzene (gray bars) at equilibrium.

**Figure 4 polymers-14-03468-f004:**
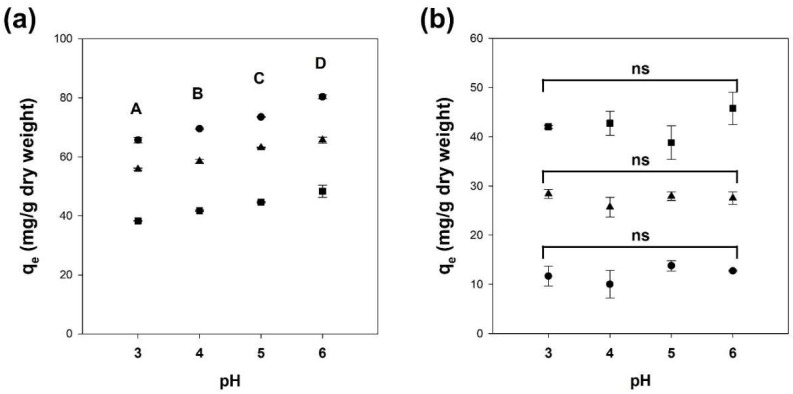
Effect of the initial solution pH on the adsorption of Cu^2+^ (**a**) and benzene (**b**) onto alginate and alginate/biochar hydrogel beads (●: 4% alginate/0% biochar, ▲: 4% alginate/1% biochar, ■: 4% alginate/4% biochar). One-way ANOVA with Tukey’s test (*p* < 0.05); ns: not significant, A, B, C, and D indicate group classified from Tukey’s test. The same letter means that there is no significant difference between the data.

**Figure 5 polymers-14-03468-f005:**
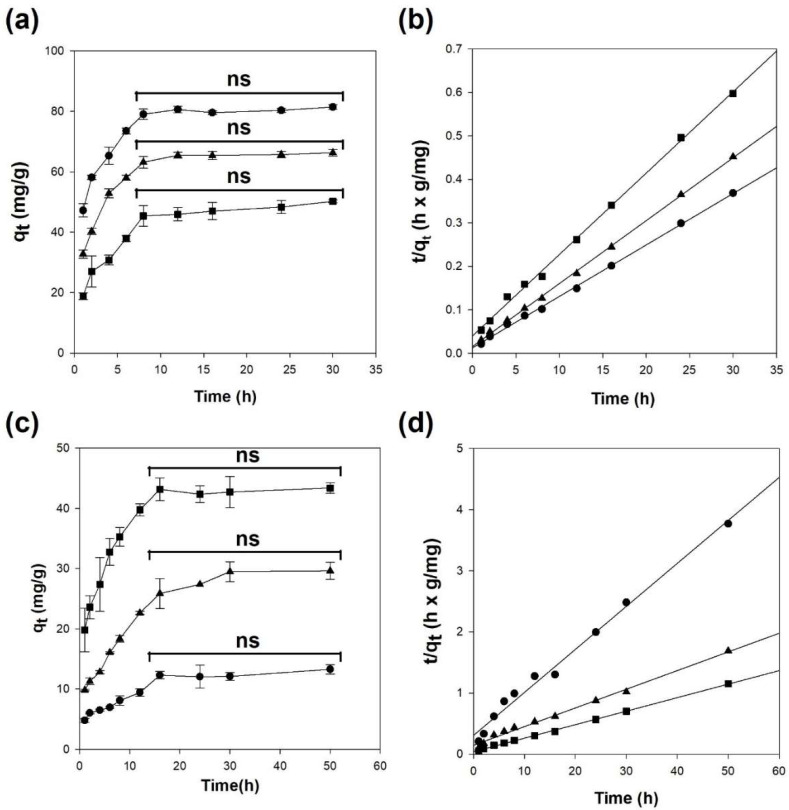
Kinetic studies of Cu^2+^ (**a**,**b**) and benzene (**c**,**d**) adsorption onto alginate and alginate/biochar hydrogel beads (●: 4% alginate/0% biochar; ▲: 4% alginate/1% biochar; ■: 4% alginate/4% biochar). Effect of contact time (**a**,**c**) and pseudo-second-order kinetic models of Cu^2+^ (**b**) and benzene (**d**). One-way ANOVA with Tukey’s test (*p* < 0.05); ns: not significant.

**Figure 6 polymers-14-03468-f006:**
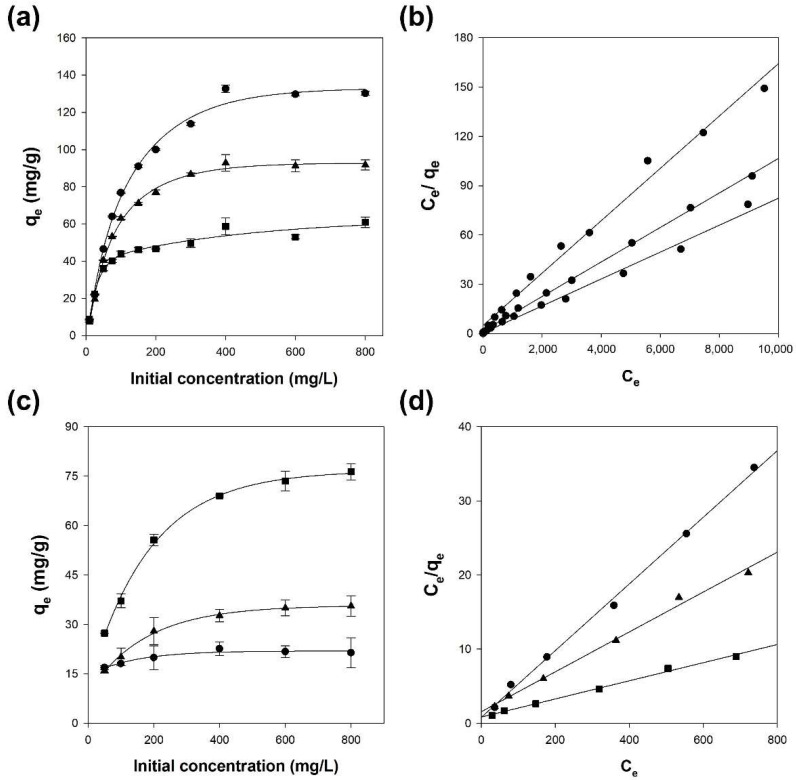
Isotherm studies of Cu^2+^ (**a**,**b**) and benzene (**c**,**d**) adsorption onto alginate and alginate/biochar hydrogel beads (●: 4% alginate/0% biochar; ▲: 4% alginate/1% biochar; ■: 4% alginate/4% biochar). Effect of initial Cu^2+^ (**a**) and benzene (**c**) concentrations and Langmuir as the best-fitted isotherm model of Cu^2+^ (**b**) and benzene (**d**).

**Figure 7 polymers-14-03468-f007:**
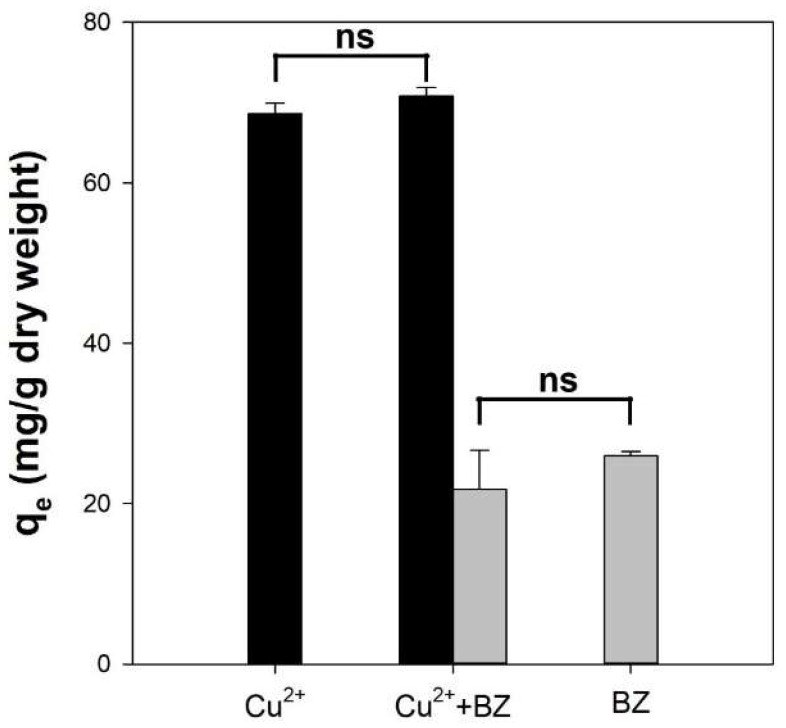
Comparison of the adsorption capacities of single contaminant Cu^2+^ (black bar) or benzene (gray bar) and contaminant mixture (Cu^2+^ and benzene) solution onto alginate/biochar hydrogel beads (4% alginate/4% biochar). *t*-test; ns: not significant.

**Figure 8 polymers-14-03468-f008:**
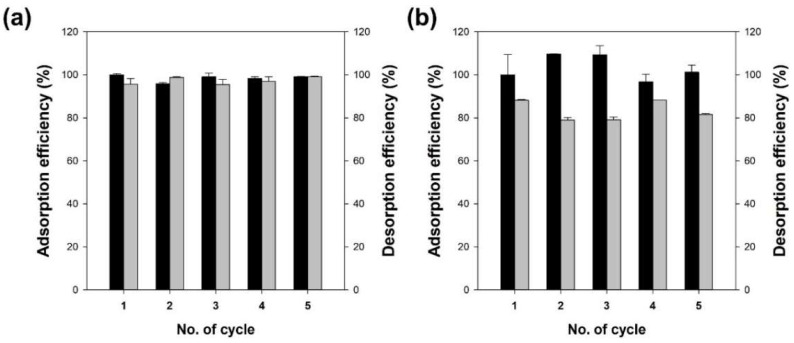
Adsorption capacities (black bars) and desorption efficiencies (gray bars) of alginate/biochar hydrogel beads (4% alginate/1% biochar) for Cu^2+^ (**a**) and (4% alginate/4% biochar) for benzene (**b**).

**Table 1 polymers-14-03468-t001:** Kinetic parameters of Cu^2+^ and benzene adsorption onto alginate and alginate/biochar hydrogel beads (1 and 4% BC).

Adsorbate	Added BC Content (%)	Pseudo-First-Order			Pseudo-Second-Order				Elovich			q_e_ Exp. (mg/g)
		k_1_ (10^−3^/h)	q_e1_ Cal. (mg/g)	r^2^	k_2_ (10^−3^g/mg/h)	q_e2_ Cal. (mg/g)	h (mg/g/h)	r^2^	a (mg/g/h)	b (g/mg)	r^2^	
Cu^2+^	0	291.6	43.6	0.855	11.3	84.5	80.8	0.999	764.1	0.088	0.902	80.2
	1	378.5	53.0	0.971	13.5	69.1	64.3	0.999	340.9	0.096	0.904	65.2
	4	292.5	40.1	0.957	8.8	53.4	25.1	0.998	79.8	0.105	0.946	47.4
Benzene	0	118.0	9.7	0.972	16.1	14.2	3.3	0.989	12.4	0.417	0.915	12.4
	1	146.0	25.9	0.981	6.4	32.7	6.8	0.992	20.0	0.166	0.944	28.1
	4	164.3	27.2	0.998	12.1	45.2	24.8	0.999	109.2	0.145	0.937	42.9

Cal: calculated; Exp: experimental; k_1_ and k_2_: rate constants of the pseudo-first-order and pseudo-second-order models, respectively; q_e1_ and q_e2_: calculated adsorption capacity at equilibrium of the pseudo-first-order and pseudo-second-order models, respectively.

**Table 2 polymers-14-03468-t002:** Isotherm parameters of Cu^2+^ and benzene adsorption onto alginate and alginate/biochar hydrogel beads (1 and 4% BC).

**Adsorbate**	**Added BC Content (%)**	**Langmuir**			**Freundlich**			
		**q_m_ (mg/g)**	**k_L_ (10^−3^L/mg)**	**r^2^**	**1/n**	**n**	**k_F_ (mg/g)**	**r^2^**
Cu^2+^	0	122.0	17.7	0.992	0.290	3.4	12.1	0.877
	1	93.5	7.2	0.999	0.285	3.5	9.1	0.861
	4	62.5	3.5	0.990	0.223	4.5	9.2	0.854
Benzene	0	22.2	56.9	0.997	0.113	8.8	10.6	0.740
	1	37.1	17.4	0.994	0.260	3.9	6.6	0.950
	4	81.7	14.7	0.995	0.321	3.1	9.9	0.958
**Adsorbate**	**Added BC Content (%)**	**Elovich**			**Dubinin** **–** **Radushkevich**			
		**q_m_ (mg/g)**	**K_E_ (L/g)**	**r^2^**	**q_m_ (mg/g)**	**k_DR_** **(10^−6^ mol^2^/kJ^2^)**	**E (kJ/mol)**	**r^2^**
Cu^2+^	0	22.6	0.34	0.922	94.3	4.9	0.32	0.859
	1	16.9	0.30	0.917	74.5	12.3	0.20	0.891
	4	9.3	0.66	0.912	46.7	18.0	0.17	0.805
Benzene	0	3.4	9.79	0.685	20.4	52.1	0.10	0.377
	1	9.1	0.29	0.930	30.7	156.8	0.06	0.803
	4	23.7	0.13	0.943	63.5	128.1	0.06	0.791

## Data Availability

Not applicable.

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
