# Peer review of "Adsorptive Behavior of Cu2+ and Benzene in Single and Binary Solutions onto Alginate Composite Hydrogel Beads Containing Pitch Pine-Based Biochar"

_polymers, 2022, doi:10.3390/polym14173468_

Round 1

Reviewer 1 Report

Manuscript ID: polymers-1883007

Title: Adsorptive behavior of Cu2+ and benzene in single and binary solutions onto alginate composite hydrogel beads containing pitch pine-based biochar

The authors described the preparation and characterization (SEM, FTIR spectroscopy, BET surface area) of alginate composite hydrogel beads containing various quantities of biochar produced from pitch pine. The composites’ adsorption properties were studied in the removing of Cu(II) and benzene from aqueous solutions. The adsorption kinetics and isotherm studies were performed. The main conclusion of the study is that the alginate/biochar hydrogel beads can be applied as adsorbents for the removal of multiple pollutants with different properties from wastewater.

The manuscript is generally well written and the conclusions are based on the experimental evidences. I have the following observations:

1.       The determination of Cu(II) is not described clearly enough. The principle of method and/or a brief description of procedure are necessary.

2.       “The benzene concentration was measured at 254 nm.” – For quantitative determination, a calibration curve is necessary. The authors must provide information on this issue.

3.       The sampling procedure during the adsorption experiment is not described in “Experimental” section.

4.       A reference is required for information in Table S1.

5.       The FTIR spectra - The coordination of the functional groups to Cu(II) cannot have the effect of the disappearance of FTIR bands, but their shifting. A lower intensity of the bands is not proper correlated with the coordination to the metal ions.

6.         “The dimensionless separation factor (RL)” must be defined and the calculation mode must be provided, with a reference.

Author Response

[Reviewer #1’s comment #1]

The determination of Cu (II) is not described clearly enough. The principle of method and/or a brief description of procedure are necessary.

[Response by author]

We really appreciate your valuable advice. As you suggested, we have now made more detailed information for the measurement of Cu2+ in experimental section. Hope our justification and clarification is satisfying to you.

[Action]

Page 3, line 146-page 4, line 149

“BSA (Bovine Serum Albumin) solution (10 mg/mL) was added into BCA reagents instead of CuSO4 solution. The pre-pared BCA working reagents were mixed with sample and incubated at 60 °C for 1h. The reacted samples were measured at 562 nm.”

[Reviewer #1’s comment #2]

“The benzene concentration was measured at 254 nm.” – For quantitative determination, a calibration curve is necessary. The authors must provide information on this issue.

[Response by author]

We really appreciate your valuable advice. As you suggested, we have now added the equation of standard curve in experimental section and the graph below for the determination of benzene. Hope our justification and clarification is satisfying to you.

[Action]

Page 4, line 149-150.

“The benzene concentration was quantified using standard curve (y = 1.508x10-3 x – 6.983x10-3, R2 = 0.999) at 254 nm.”

[Reviewer #1’s comment #3]

The sampling procedure during the adsorption experiment is not described in “Experimental” section.

[Response by author]

We really appreciate your valuable advice. As you suggested, we have now made more detailed information for sampling procedure in experimental section. Hope our justification and clarification is satisfying to you.

[Action]

Page 3, line 139-143

“After each adsorption reaction, for the measurement of Cu2+, aliquots (200 mL) were periodically collected and centrifuged before dilution with DW. To determine the benzene concentration, aliquots (1 mL) were filtrated using PVDF syringe filter (0.22 mm). Then, the remaining Cu2+ and benzene concentration in samples were analyzed.”

[Reviewer #1’s comment #4]

A reference is required for information in Table S1.

[Response by author]

We really appreciate your valuable advice. As you suggested, we have now checked and added references in supplementary information. Hope our justification and clarification is satisfying to you.

[Action]

References ([1] and [2]) were added to Table S1.

[Reviewer #1’s comment #5]

The FTIR spectra - The coordination of the functional groups to Cu(II) cannot have the effect of the disappearance of FTIR bands, but their shifting. A lower intensity of the bands is not proper correlated with the coordination to the metal ions.

[Response by author]

We really appreciate your valuable advice. As you suggested, we have now revised the sentences in results and discussion section. Hope our justification and clarification is satisfying to you.

[Action]

Page 7, line 262-266

“In addition, there were no significant changes of various peaks before and after Cu2+ and benzene adsorption. Nevertheless, some changes (shift and/or intensity) may occur after Cu2+ adsorption as reported form the previous literature [42]. Mohammed et. al. (2018) reported that the interaction between the divalent ion (e.g., Pb2+, Cu2+, and Cd2+) and carboxylate group leads the ability to bind [42].”

[Reviewer #1’s comment #6]

“The dimensionless separation factor (RL)” must be defined and the calculation mode must be provided, with a reference.

[Response by author]

We really appreciate your valuable advice. As you suggested, we have now made more detailed information for the dimensionless separation factor (RL) in result section. And we indicated the equation of RL in the supplementary information (Table S1). Hope our justification and clarification is satisfying to you.

[Action]

Page 11, line 370-373

“In the Langmuir equation, the Langmuir constant (KL) indicates surface area and porosity correlated with adsorption capacity of adsorbents. Thus, the dimensionless separation factor (RL) acquired from the Langmuir model means the favorability of adsorption.”

Reviewer 2 Report

Wastewater treatment is important for the sustainable development. Biomass and their derived biochars are promising materials for wastewater treatment attributing to their low cost and high specific surface area. The research of this manuscript is interesting and results are reliable. However, major revision is required and the comments are given below.

1.     It would be better to write the units in the same way. “cm-1”, “mg/mL” and other units need to be revised.

2.     Wastewater treatment is a hot topic and many absorbents with good performance are developed. It would be better to describe different absorbents in the introduction part for broad readers. More references are suggested to be cited, especially those newly published. Please refer and cite: A review on conversion of crayfish-shell derivatives to functional materials and their environmental applications; Synthesis and Application of Granular Activated Carbon from Biomass Waste Materials for Water Treatment: A Review; MOFs meet wood: Reusable magnetic hydrophilic composites toward efficient water treatment with super-high dye adsorption capacity at high dye concentration.

3.     What does the “A”, “B” and “C” in Figure 1 means? The same question to Figure 4a.

4.     The scale bars for SEM images in Figure 2 and Figure S2 are hard to tell. Please add more clear ones.

5.     “BET surface area” should be revised as “BET specific surface area”.

6.     It is very interesting that BET specific surface area of pitch pine-based biochar produced at 550 and 750 °C shows such large difference. Could the authors give some possible reasons? Moreover, the nitrogen absorption/desorption isothermal curves and pore size distribution curves are suggested to be added.

7.     The authors should pay attention to the writing of references, especially the journal names. The writing of metal ions in reference 39 need to be revised.

Author Response

[Reviewer #2’s comment #1]

It would be better to write the units in the same way. “cm-1”, “mg/mL” and other units need to be revised.

[Response by author]

We really appreciate your valuable advice. As you suggested, we have now made all units in the same way (e.g., mg/mL) except of cm-1 due to the general unit of FTIR. Hope our justification and clarification is satisfying to you.

[Reviewer #2’s comment #2]

Wastewater treatment is a hot topic and many absorbents with good performance are developed. It would be better to describe different absorbents in the introduction part for broad readers. More references are suggested to be cited, especially those newly published. Please refer and cite: A review on conversion of crayfish-shell derivatives to functional materials and their environmental applications; Synthesis and Application of Granular Activated Carbon from Biomass Waste Materials for Water Treatment: A Review; MOFs meet wood: Reusable magnetic hydrophilic composites toward efficient water treatment with super-high dye adsorption capacity at high dye concentration.

[Response by author]

We really appreciate your valuable advice. As you suggested, we have now added the suggested references in introduction section. Hope our justification and clarification is satisfying to you.

[Action]

Page1, line 43-page2, line 46

“In particular, biomass derivatives are one of the candidates considered as alternative materials for conventional adsorbents owing to their recyclable, biodegradable, and available features. [5-7].”

[Reviewer #2’s comment #3]

What does the “A”, “B” and “C” in Figure 1 means? The same question to Figure 4a.

[Response by author]

We really appreciate your valuable advice.” A, B, C, and D” indicate group classified from Tukey’s test. The same letter means that there is no significant difference between the data.

[Action]

Page5, line 223-224

A, B and C indicate group classified from Tukey’s test. The same letter means that there is no significant difference between the data.

[Reviewer #2’s comment #4]

The scale bars for SEM images in Figure 2 and Figure S2 are hard to tell. Please add more clear ones.

[Response by author]

We really appreciate your valuable advice. As you suggested, we have now revised the figures. Hope our justification and clarification is satisfying to you.

[Action]

Page 6, line 238, Figure 2.

Supplementary information, Figure S2.

[Reviewer #2’s comment #5]

“BET surface area” should be revised as “BET specific surface area”.

[Response by author]

We really appreciate your valuable advice. As you suggested, we have now revised. Hope our justification and clarification is satisfying to you.

[Action]

Page 3, 123; page 14, 453; Supplementary information, Table S2.

[Reviewer #2’s comment #6]

It is very interesting that BET specific surface area of pitch pine-based biochar produced at 550 and 750 °C shows such large difference. Could the authors give some possible reasons? Moreover, the nitrogen absorption/desorption isothermal curves and pore size distribution curves are suggested to be added.

[Response by author]

We really appreciate your valuable advice. As you suggested, we have now added the sentences in results and discussion section for reason of deceased BET specific surface area at 750 °C compared to it at 550 °C. Also, we have added the isothermal curves in here. Hope our justification and clarification is satisfying to you.

[Action]

Page 4, line 191-page 5, line 196

“The surface area of the biochar produced at 550 °C was larger than that produced at 350 °C, whereas the biochar produced at 750 °C had a smaller surface area than that produced at 550 °C (Table S2). These phenomena may indicate that high temperatures can break down the porous structures of biochar and plug the pores with tar produced through the biomass pyrolysis process, leading to a decrease in the surface area of the biochar.”

For 350 °C (Please check the attached file.)

For 550 °C (Please check the attached file.)

For 750 °C (Please check the attached file.)

[Reviewer #2’s comment #7]

The authors should pay attention to the writing of references, especially the journal names. The writing of metal ions in reference 39 need to be revised.

[Response by author]

We really appreciate your valuable advice. As you suggested, we have now revised references. Hope our justification and clarification is satisfying to you.

Round 2

Reviewer 2 Report

The manuscript is well revised according to the comments and is acceptable.